# Brolucizumab in Pretreated Neovascular Age-Related Macular Degeneration: Case Series, Systematic Review, and Meta-Analysis

**DOI:** 10.3390/life13030814

**Published:** 2023-03-17

**Authors:** Christof Hänsli, Christin Schild, Isabel Pfister, Justus G. Garweg

**Affiliations:** 1Berner Augenklinik, 3007 Bern, Switzerland; 2Department of Ophthalmology, Faculty of Medicine, University of Bern, 3012 Bern, Switzerland; 3Department of Ophthalmology, Inselspital, Bern University Hospital, University of Bern, 3012 Bern, Switzerland

**Keywords:** neovascular age-related macular degeneration, brolucizumab, switch, meta-analysis, anti-VEGF, recalcitrant nAMD, best-corrected visual acuity, central subfield thickness, long-term follow-up

## Abstract

Background: Recalcitrant neovascular age-related macular degeneration (rnAMD) despite intensive intravitreal anti-neovascular endothelial growth factor (VEGF) treatment, can be handled by switching to another anti-VEGF agent. This first systematic review and meta-analysis presents long-term data after switching from another anti-VEGF agent to brolucizumab. Methods: Retrospective case series over two years of patients switched to brolucizumab, and a systematic review and meta-analysis of peer-reviewed studies presenting patients switched to brolucizumab. Weighted mean differences based on the random-effects models were calculated for best-corrected visual acuity (BCVA) and central subfield thickness (CST). Results: The systematic review draws on 1200 eyes switched to brolucizumab. The meta-analysis showed a clinically irrelevant decrease in BCVA after one and two months, together with significant decreases in CST for up to one year after the switch but lacking power over 2 years. Of twelve eyes (twelve patients) in our case series, five continued treatment for two years without experiencing significant changes. Conclusions: After switch to brolucizumab, a significant morphological improvement with CST reduction was shown in eyes with rnAMD. The small worsening of BCVA may be owing to the chronically active nature of rnAMD. Brolucizumab thus remains a treatment option in rnAMD despite its potential side effects.

## 1. Introduction

Age-related macular degeneration (AMD) is the main cause of visual impairment and blindness in the aging population of Western countries, with an incidence of 2.4% in individuals beyond 60 years [1]. Neovascular age-related macular degeneration (nAMD) is a treatable form of advanced AMD. Anti-vascular endothelial growth factor treatment (anti-VEGF) enables improvement of best-corrected visual acuity (BCVA) and disease stabilization with initially monthly intravitreal treatment (IVT) [2]. Though a correlation of intra- or subretinal fluid with long-term functional outcomes, i.e., best-corrected visual acuity (BCVA), has been demonstrated [3], the high treatment burden for both patients and caregivers necessitates a continuing search for new drugs and treatment regimens. To date, in the USA and Europe, ranibizumab (Lucentis^®^, Novartis, Switzerland), aflibercept (Eylea^®^, Bayer, Germany), brolucizumab (Beovu^®^, Novartis, Switzerland), and faricimab (Vabysmo^®^, Roche, Switzerland) are licensed for the treatment of nAMD. Bevacizumab (Avastin^®^, Roche, Switzerland), ZIV-Aflibercept, and conbercept (Lumitin^®^, Chengdu Kanghong, China) are also frequently used in many parts of the world. To reduce the treatment burden, various protocols have been established, such as pro-re-nata (PRN) [4] and treat and extend (TAE) [5], and various modifications such as observe and plan [6]. With the newer anti-VEGF agents, a majority of patients experience disease stabilization, with injections needed every three months after the first year [7]. 

However, a relevant percentage of patients remains with a high treatment demand due to recalcitrant or recurrent intra- and/or subretinal fluid despite regular injections of anti-VEGF in monthly or six-weekly intervals. In those cases of pre-treated recalcitrant neovascular AMD (rnAMD), a switch to another anti-VEGF agent has shown the potential to improve morphology and treatment demand, although with minor effects on vision in the short-term [8,9]. The development of new and potentially longer-lasting anti-VEGF agents sparked hope that in high treatment demand, a switch to newer products might be beneficial for both macular morphology and function. The phase 3 clinical trials HAWK and HARRIER proved non-inferiority with regard to BCVA and indicated an improved reduction of central subfield thickness (CST) with brolucizumab every eight to twelve weeks compared to aflibercept given eight-weekly [10]. Early reports showed a reduction of intraretinal fluid and CST in eyes with high treatment demand that switched to brolucizumab [11,12]. On the other hand, with the increased use of brolucizumab upon approval by the U.S. food and drug agency (FDA), new adverse events surfaced. Eyes treated with brolucizumab experienced higher rates of intraocular inflammations (IOI) compared to other anti-VEGF agents. Some of the IOI led to occlusive retinal vasculitis (ORV) with severe, irreversible visual loss [13,14]. This prompted ophthalmologists to move away or switch back from brolucizumab [15]. Despite this fact, the overall non-inferiority of brolucizumab compared to aflibercept with regard to mean BCVA change remained [14]. Significant efforts were put into understanding the pathophysiology and preventive measures of such severe side effects. With its potentially higher efficacy, however, brolucizumab remains a treatment option in cases of high treatment demand. To date, MERLIN remains the only prospective randomized trial that has been published on this topic so far [16]. However, the trial was terminated early due to the per-protocol continuous, every-4-weeks dosing up to week 52, going along with higher rates of IOI including retinal vasculitis and retinal vascular occlusions as compared to aflibercept. FALCON is an ongoing randomized clinical trial comparing the switch to brolucizumab in eyes with rnAMD with or without three initial monthly loading doses [17]. Based on rigid inclusion criteria, it has shown difficulties recruiting the planned target sample sizes. 

Thus, current evidence for the role of brolucizumab in pre-treated eyes with a high treatment demand is grounded on an increasing number of small retrospective studies and case series. Due to the known benefits of anti-VEGF agent switching [8,9], a focused analysis on brolucizumab is worthwhile. In the absence of prospective evidence, this systematic review and meta-analysis summarizes current published and own experience in pre-treated eyes with nAMD that were switched to brolucizumab. 

## 2. Materials and Methods

### 2.1. Case Series

#### 2.1.1. Inclusion and Exclusion Criteria for Case Series

In our own consecutive single-center case series, pre-treated eyes that were switched from ranibizumab or aflibercept to brolucizumab due to a high treatment demand for rnAMD between February and May 2020 were included. Patients had a follow-up of at least two years after the switch. This case series is a follow-up of the previously published 6-month results [18]. Patients with nAMD and persistent or recurrent intra- or subretinal fluid despite four- to six-weekly intravitreal anti-VEGF (aflibercept or ranibizumab) treatment under a TAE regimen were offered a switch to brolucizumab with the aim to reduce the treatment burden. Patients were informed about both the potential benefits and risks of this drug before they agreed to switch to brolucizumab. This study adhered to the tenets of the 1964 Declaration of Helsinki and its later amendments and was approved by the local institutional ethical committee (Kantonale Ethikkommission Bern, BASEC-ID 2020-00412). Upon inclusion, patients received brolucizumab injections in a TAE regimen without a loading phase. At the discretion of the treating physician, treatment with brolucizumab could be discontinued according to medical needs. For these patients, follow-up continued during two years. At every visit, a thorough clinical investigation including dilated fundus examination was performed. Best-corrected distance visual acuity (BCVA) and best-corrected reading acuity (RA) were assessed with standard Snellen decimal charts for distance and Radner reading charts for near with additional +3.0 diopters. Visual acuity was converted to the logarithm of the minimum angle of resolution (LogMAR) and reading acuity to the equivalent logarithmic reading acuity determination (LogRAD) for analysis. CST was measured with spectral-domain optical coherence tomography (OCT, Heidelberg Engineering, Heidelberg, Germany), where manual correction of segmentation was performed as needed. Injection burden was assessed as number of injections per time and treatment intervals. Findings at switch (baseline) and after 12 and 24 months were compared. 

#### 2.1.2. Statistical Methods for Case Series

Patients who continued brolucizumab treatment over 2 years were analyzed separately from those who discontinued treatment for complications or switched back to their previous anti-VEGF. Mean values for BCVA and CST were compared from baseline to 12 and 24 months. Normal distribution was tested with Shapiro–Wilk test. Differences were tested with the paired-samples *t*-test for normally distributed continuous data or Wilcoxon signed-rank in case of not normally distributed data. For group comparisons of continuous data, we applied a *t*-test for normally distributed data or Mann–Whitney U-test for not normally distributed data. Data are presented as mean ± standard deviation (SD). A *p* ≤ 0.05 was considered as statistically significant. Statistical analyses were performed using the SPSS software package 28.0.1 (SPSS, Inc., Chicago, IL, USA) and R (version 3.2.4; R: A language and environment for statistical computing, R Foundation for Statistical Computing, Vienna, Austria, 2016). 

### 2.2. Systematic Review and Meta-Analysis

#### 2.2.1. Inclusion and Exclusion Criteria for Meta-Analysis

The literature search and analysis were performed according to the preferred reporting items for systematic reviews and meta-analyses (PRISMA) strategy 2020 and checklist [19]. A systematic literature search was performed on 29 August 2022 in the *NCBI*/*PubMed* database from the National Institutes of Health (USA), *ScienceDirect*, the *Cochrane Central Register of Controlled Trials* (*CENTRAL*), *Google Scholar*, and *Scopus* to identify any pro- and retrospective study retrieved by the key terms (brolucizumab, beovu, aflibercept, bevacizumab, ranibizumab, and anti-VEGF). The search strategies are presented in Appendix A. Further, reference lists of published reports, meta-analyses, and reviews were screened for suitable original articles. All retro- and prospective studies were included that provided data on eyes with pre-treated nAMD that were switched to brolucizumab, with a minimal follow-up time of one month, and were written in English, French, or German. Papers were not included if one of the following criteria was met: single case reports as well as case series of complications for reporting bias; presentations on conferences; reviews; reporting and meta-analyses summarizing elsewhere reported patients and observations; editorial notes not reporting new observations. Additional exclusion criteria were not applied in order to prevent any bias. By including only peer-reviewed journals, the dataset ensures an appropriate level of methodological robustness. In general, an inherent bias cannot be avoided based on the partially small number of observations per paper. All references were managed using Zotero open-source referencing software (Version 6.0.21). Automatic check for duplicates was conducted as well as manual screening.

The selection process was conducted by two researchers (C.H. and C.S.) using a stepwise approach including title and abstract screening. Differences were resolved by discussion. All resulting manuscripts were evaluated by the authors based on the above-mentioned in- and exclusion criteria. One author extracted the data from all suitable references in a two-step process: first, all necessary information based on a coding sheet draft was entered. Additional categories were then added to the coding scheme according to the data presented in the papers. In the second step, missing data were specifically searched in the research papers. All data entries were confirmed by the first author. 

#### 2.2.2. Statistical Methods for Meta-Analysis

The following information was extracted in mean with standard deviation (as far as available): best-corrected visual acuity (BCVA) and central subfield thickness (CST) measured with optical coherence tomography. The included timepoints were the time of the switch, one month after injection, after four to six months (pooled), after one year, and after 18 to 24 months (pooled). Reported complications were collected per study. In general, we collected as much data as possible to provide a comprehensive overview on published cases. In some instances, the unit of data was transformed to enable synthesis, including visual acuity (transformed to ETDRS letters) and durations (months, weeks, days). 

For a demographic overview of our studies, we calculated weighted mean values for the demographics age and gender ratio and a frequency distribution for the country of origin. For the meta-analytic integration of results, we calculated weighted mean differences based on the random-effects models using comprehensive meta-analysis (CMA) software Version 3.0 [20] for visual acuity scores (ETDRS) and CST at baseline and one 4 to 6 months, 12, and 18 to 24 months of follow-up. Studies reporting absolute mean BCVA and CST values with standard deviations (SD) were included in the meta-analysis. Single reporting of median with ranges or interquartile ranges were excluded. BCVA was either reported or converted into Early Treatment of Diabetic Retinopathy Study Letter Score (ETDRS, with 85 ETDRS letters accounting for a visual acuity of 1.0 Snellen or 0 LogMAR). We used the subgroup within study-level as the level of analysis, meaning that every subgroup acted as an independent sample as opposed to pooling them. In case of insufficient data reporting, we could not convert values into absolute measurements and therefore had to exclude the studies from the meta-analytic calculations. To avoid missing a higher number of papers due to insufficient data reporting, we also calculated weighted mean values for every time point. For this calculation, all available data have been integrated. As a consequence, numbers can differ between meta-analytic calculations and weighted mean values. 

## 3. Results

### 3.1. Case Series

#### 3.1.1. Patients

Between February and May 2020, twelve eyes of twelve patients with nAMD and high treatment demand were switched to brolucizumab and followed-up until June 2022. Seven of the twelve eyes (58.3%) discontinued treatment. Two developed intraocular inflammation after one and two brolucizumab injections, respectively. Of these, one eye had intermediate uveitis and one eye anterior uveitis with extramacular occlusive retinal vasculitis, which were treated with topical and systemic corticosteroids [18]. One additional eye developed ocular ischemic syndrome after six months and five brolucizumab injections, and one eye showed insufficient response with persistent intraretinal fluid and was switched back to aflibercept after one year. One patient was lost to follow-up. One patient with a history of stroke experienced a transient ischemic attack after three brolucizumab injections. Although it was not presumed to be caused by intravitreal brolucizumab, treatment was discontinued and five months later resumed with aflibercept as a precaution measure. Finally, one eye continued to have high treatment demand with brolucizumab and thus was administered an intravitreal long-release dexamethasone implant (Ozurdex^®^, AbbVie Inc., North Chicago, IL, USA) according to the clinic’s previous experience (unpublished data). To avoid a reporting bias, these patients were analyzed together as discontinuation group (DISC), allowing comparability within an intention-to-treat approach. The remaining five patients that continued brolucizumab treatment as intended were analyzed as the brolucizumab group (BROL).

#### 3.1.2. Intravitreal Injections

Baseline characteristics were comparable in patients in the BROL and DISC groups. Over the last 12 months prior to the switch, patients had received 10.0 ± 1.4 (BROL) and 10.3 ± 1.0 anti-VEGF injections (DISC), with a last pre-switch interval of 5.2 ± 1.1 and 4.7 ± 0.8 weeks, respectively (*p* = 0.34). The total number of intravitreal injections (IVI) was 40.8 ± 28.3 over 111.5 ± 34.1 months in BROL and 27.3 ± 16.8 over 35.2 ± 26.1 months in DISC. The BROL group received 7.8 ± 1.8 brolucizumab injections in the first year and a total of 13.2 ± 3.5 IVI over two years. The discontinuation group received 8.0 ± 1.2 IVT in the first year and a total amount of 12.5 ± 4.0 IVI over two years. The last treatment interval in the BROL group was 11.0 ± 4.0 weeks. In the DISC group, the last interval was 19.0 ±16.4 weeks, including patients with discontinued treatments for various reasons. No significant difference in the amount of IVI was detected between the two groups. 

#### 3.1.3. Visual Acuity

Mean BCVA remained stable and comparable between the groups during the follow-up. In detail, BCVA at baseline was 0.32 ± 0.15 logMAR for BROL and 0.53 ± 0.27 logMAR for DISC and did not differ between the two groups (*p* = 0.12). No significant difference in BCVA between the groups was detected over the course of the study. For BROL, BCVA after 12 and 24 months was 0.20 ± 0.12 logMAR and 0.30 ± 0.21 logMAR, respectively, without significant differences compared to baseline (paired *t*-test for 12 months: *p* = 0.071, for 24 months: *p* = 0.74, Figure 1a). 

#### 3.1.4. Reading Acuity

Mean RA at baseline was comparable between the groups (BROL: 0.44 ± 0.17; DISC: 0.60 ± 0.26; *p* = 0.12) and remained stable over 12–24 months in the BROL group, while no RA outcomes were recorded in the DISC group. For the BROL group, RA remained stable over 12 and 24 months with 0.30 ± 0.19 and 0.40 ± 0.20, without differences compared to baseline (paired *t*-test for 12 months: *p* = 0.051 for 24 months: *p* = 1.0). 

#### 3.1.5. Central Subfield Thickness

Central subfield thickness was similar between the groups at baseline and at all time points thereafter. In detail, CST at baseline was 394.2 ± 100.7 µm in BROL and 562.0 ± 204.4 µm in DISC. There was no significant difference between the groups at any time point. In the BROL group, CST at 12 months was 320.0 ± 40.3 µm and after 24 months 364.6 ± 93.9 µm. At 12 and 24 months, no statistically significant difference compared to baseline could be detected (Wilcoxon signed rank test, *p* > 0.05; Figure 1b).

### 3.2. Review and Meta-Analysis

#### 3.2.1. Information on Included Studies

After the literature search, of 699 publications 202 duplicates were removed (see PRISMA Search Flow, Figure 2). Screening of the remaining 497 titles excluded another 347 papers. Of 150 reviewed abstracts, 96 were excluded, mainly studies with only treatment-naïve eyes, case reports, or opinion articles. The 54 full-text screenings revealed 16 peer-reviewed publications (including one published correction) matching the in- and exclusion criteria and providing sufficient data for analysis. By additional screening of cross-references, four papers were identified and included as well. This added to a total of 20 peer-reviewed publications (including one published correction) representing 20 samples within 18 independent studies. Further, we added our above-presented case series to the analysis, which extends the data for one of the included publications [18]. 

Most papers reported short-term outcomes for one month (k = 13 cohorts, n = 986 eyes for CST, k = 12 cohorts, n = 627 eyes for BCVA) or four to six months (k = 8 cohorts and n = 182 eyes for CST, k = 7 cohorts and n = 161 eyes for BCVA). One-year data were published for five cohorts with 468 eyes for CST and four cohorts with 422 eyes for BCVA. Longer-term data of 18 to 24 months were only available from our sample of five eyes for CST and for BCVA from three cohorts including a total of 47 eyes.

In total, we included 1200 eyes with a mean follow-up period of 33.4 weeks (n = 927 eyes). The reported age was mean 76.6 ± 7.7 years (n = 1113), with 49.6% female patients (n = 1012). The analysis includes studies from the USA (n = 681 eyes), Germany (n = 185 eyes), India (n = 95 eyes), Japan (n = 90 eyes), Canada (n = 73 patients), Austria (n = 36 eyes), Switzerland (n = 31 eyes), and Italy (n = 9 eyes).

Reported duration of follow-up after the switch to brolucizumab was 38.5 ± 4.7 weeks (n_mean_ = 703, n_SD_ = 216), with a mean of 5.9 ± 1.5 brolucizumab injections applied (n_mean_ = 1038, n_SD_ = 588). The final reported treatment interval after multiple injections was 9.3 ± 2.7 weeks (n = 148).

#### 3.2.2. Visual Acuity

At the time of the switch, weighted mean BCVA was 63.3 ± 11.9 ETDRS (n_mean_ = 1172, n_SD_ = 1100). One month after the first intravitreal injection of brolucizumab, weighted BCVA was 61.5 ± 16.9 ETDRS (n_mean_ = 627, n_SD_ = 576). Four to six months after switch, weighted BCVA was 62.4 ± 15.8 ETDRS (n_mean_ = 161, n_SD_ = 161). At the one-year follow-up, weighted BCVA increased to 72.1 ± 16.5 ETDRS (n_mean_ = 323, n_SD_ = 26), and finally, 18–24 months after the switch to brolucizumab, weighted BCVA was 77.1 ± 8.3 ETDRS (n_mean_ = 47, n_SD_ = 47; see Figure 3a). 

The differences between switch and the four time points were been meta-analyzed (Table 1). The effect size (point estimate) for the difference between the switch and 1 month follow-up is 0.132, which equals a small but stable effect (*p* = 0.025). For the 4- to 6-month follow-up, the effect is 0.295, equaling a small but significant effect (*p* = 0.006). Both the differences between the switch and 12 months and 18 to 24 months of follow-up reached a small- to medium-sized effect but failed to reach significance. It is important to note that only a subsample of the above-presented weighted values were meta-analyzed. Only studies presenting mean and standard deviation could be included. 

#### 3.2.3. Central Subfield Thickness

At the time of the switch, weighted mean CST was 372.1 ± 96.6µm (n_mean_ = 1130, n_SD_ = 1079). One month after the first intravitreal injection of brolucizumab, weighted CST was 308.0 ± 65.4 µm (n_mean_ = 986, Δ = −17.3% n_SD_ = 582). Four to six months after the switch, weighted CST was 299.1 ± 72.6 µm (n_mean_ = 182, n_SD_ = 182). After one year, weighted CST was 322.5 ± 42.4 µm (n_mean_ = 468, n_SD_ = 171). Finally, 18–24 months after the switch to brolucizumab, weighted CST was 364.6 ± 93.3 µm (n_mean_ = 5, n_SD_ = 5, Figure 3b). 

For CST, we also meta-analyzed the differences between the switch and the follow-ups (Table 1). For the difference between the switch and 1-month follow-up, we observed a negative effect (d = −0.815, *p* < 0.001), and the effect tended to be larger after 4–6-month follow-up (d = −1.027, *p* < 0.001) and further increased until one year after the switch (d = −1.800, *p* = 0.005). This demonstrates both a relevant and significant decrease of the CST after the switch to brolucizumab for up to one year. At the two-year follow-up, this effect was still present, but due to the small number of observations, it failed to reach significance. As for the calculations for visual acuity, we only included studies presenting mean and standard deviation in this analysis.

#### 3.2.4. Treatment Intervals

Three papers with four subgroups reported change of treatment intervals: Ueda-Consolve et al. reported an increase from 7.4 ± 1.4 weeks to 11.6 ± 2.6 weeks after the switch in 19 eyes with type 1 macular neovascularization and from 6.9 ± 1.3 weeks to 11.7 ± 3.1 weeks after the switch in 23 eyes with polypoidal choroidal vasculopathy after 18 months [21]. Giunta et al. showed an increase from 4.7 ± 2.3 weeks to 6.8 ± 2.8 weeks in a cohort of 73 patients at least three months after the switch to brolucizumab [22]. Our case series showed an increase from 5.2 ± 1.1 weeks before the switch to a last treatment interval of 11.0 ± 4.0 weeks two years after the switch.

## 4. Discussion

This first systematic review and meta-analysis focuses on previously treated eyes with nAMD that were switched to brolucizumab. Our meta-analysis shows that a switch to brolucizumab reduces CST in eyes with recalcitrant nAMD, a prognostic biomarker for decreased disease activity. The observed effects increased and remained significant for up to one year after the switch. The effect at the two-year follow-up lacked statistical power, with only five patients from our case series available for analysis. BCVA showed a clinically irrelevant but significant reduction in BCVA until six-month follow-up. In the analyzed subset of pretreated eyes with rnAMD, this indicates a relative stabilization of BCVA compared to a possible further decrease in case of continuing the previous anti-VEGF therapy, while evidence for this tenet is lacking. However, our analysis could not show a durable stabilization of BCVA one and two years after the switch to brolucizumab, which is attributed to a too-short follow-up in most reported series. 

Reasons to switch to brolucizumab were persistent disease activity with intra- and/or subretinal fluid and/or a high treatment demand with other anti-VEGF agents. This represents a highly specific but relevantly large subgroup of patients with nAMD [7] and demands separate analysis in contrast to pooled samples with treatment-naïve eyes, which have been widely published in both case series and meta-analyses [23]. From two prospective clinical trials addressing the same questions, one was aborted after one year, and another faced relevant recruiting issues [16,17]. Three of the included papers with four subgroup samples reported longer treatment intervals with a doubled interval on a continued TAE after the switch to brolucizumab. Despite this effect, these papers also did not report improvements of BCVA in these patients. Our findings support the clinical relevance of reducing treatment burden thanks to the improved anatomical effects of brolucizumab compared to aflibercept, which were demonstrated in HAWK and HARRIER [10,24]. The data with regard to treatment intervals were not consistently reported and therefore not included in our quantitative meta-analysis. Especially in retrospective series, treatment intervals suffer relevant bias and need to be interpreted carefully. For example, with the switch to brolucizumab, in some of the presented series, a shorter treatment interval was introduced during the loading phase according to the official label of brolucizumab. Therefore, some of the beneficiary effects may be owed to shorter treatment intervals during a loading phase and not necessarily higher drug effectiveness. On the other hand, Awh et al. showed an increased treatment interval ≥6 weeks with brolucizumab in a proportion of eyes with particularly short intervals <5 weeks before the switch [25]. 

It is important to underline that in this meta-analysis, the cohort size and therefore the power of the study was limited beyond six months of follow-up. This specifically applies for the effects after 12 to 24 months of follow-up, which failed to show significance for visual acuity. Interestingly, the point estimate of BCVA change in the two-year data shows improved visual acuity. This may be due to the over-representation of patients with type one macular neovascularization and polypoidal choroid vasculopathy (PCV), which has been reported to be particularly susceptible to brolucizumab treatment [26]. Since brolucizumab was released for the treatment of nAMD in most countries by the end of 2019 or in 2020, we can expect more reports with 24-month follow-ups to be published within the next years. With an increased number of samples in the future, the reported effects may become more robust for the 12- and 24-month follow-up data as well. Furthermore, inherent to the retrospective nature of the available studies included, the different cohorts and reported data are heterogenous. Analyzing weighted mean differences based on the random-effects models in our pooled meta-analysis allowed to increase the sample size of our long-term systematic review. However, for some studies, reported data were insufficient to be included into the meta-analysis. 

Intraocular inflammation (IOI) is a potential side effect of all anti-VEGF agents. With brolucizumab, it can result in severe vision loss in the case of occlusive retinal vasculitis. This specific side effect has been discussed with regard to incidence, treatment, outcome, and prevention in various other studies and was not in the scope of this study [13,27,28]. Reporting of side effects was not always specified for the subgroups with or without pretreatment in all studies, which prevented a quantitative exploitation. The reported incidences of intraocular inflammation under brolucizumab treatment reached from none in five studies to 22% in two studies. Occlusive vasculitis was reported in half of the studies, with a common incidence of 1–2% and a single report of 8.3% in a study with a relatively small sample size (Table 2). This falls within the wide variations of published real-life reports but is slightly higher compared to the phase 3 trials including treatment-naïve eyes [10,13].

Our own case series is one of the few reporting a long-term follow-up after the switch to brolucizumab in rnAMD and, to our knowledge, the only to attempt a two-year follow-up analysis. However, it consists of a small sample with a considerably high dropout rate. While some of the dropouts were not drug-related, we still had a higher rate of intraocular inflammation and retinal ischemic complications compared to the literature. While the initial six-month report indicated a promising effect with regard to central subfield thickness reduction and some improvement of reading acuity [18], the effects did not persist over one and two years. 

## 5. Limitations

While this is the first meta-analysis specifically focusing on previously treated nAMD with the longest available follow-up periods, some limitations need to be considered. First, all included studies were retrospective case series with various inclusion and exclusion criteria as well as different treatment protocols and reported outcome parameters, introducing some selection, reporting, and treatment bias. In addition, we decided to exclude single cases reported in the early time after brolucizumab approval as well as case series of complications where a relevant reporting bias was evident. Second, we included patients with nAMD and PCV, which are not clearly distinguished in all studies due to their similarities in appearance and treatment. However, their respective response to treatment may differ slightly. Third, we isolated data from previously treated nAMD from the published cohorts, which also included treatment-naïve eyes, but some parameters such as inflammatory side-effects were not separately disclosed with regard to baseline criteria. Fourth, the sample size decreases significantly for follow-ups of one year and more after the switch, reducing the statistical power of these time points. Finally, the data presentation of the included studies has to be considered. Only a subgroup of studies reported basic statistics such as mean and standard deviation for different time points and could be included for our meta-analytic calculations. To sum up all available data, the weighted mean calculations show a trend towards a greater dataset.

## 6. Conclusions

Our meta-analysis of pretreated eyes with nAMD switched to brolucizumab shows a relevant reduction of CST for up to one year after the switch, while BCVA remained relatively stable with a clinically irrelevant yet significant decline over six months. The anatomic response after the switch confirms a better effectiveness of brolucizumab compared to other anti-VEGF agents in cases of recalcitrant nAMD with high treatment demand. Though visual gain was not consistently observed, this is important since the presence of intra- and subretinal fluid represents a well-accepted prognostic biomarker for poor long-term visual outcomes in nAMD. The inconsistency of the reported functional responses after switching to brolucizumab in the current literature is likely explained by the selection bias of eyes with rnAMD, which demonstrate less potential for visual improvement. Such eyes have longer pre-treatment periods and have sustained intra- and subretinal fluid for longer times compared to treatment-naïve eyes. In short, brolucizumab remains an option for recalcitrant nAMD, where a reduction of the treatment burden is warranted in order to maintain BCVA.

## Figures and Tables

**Figure 1 life-13-00814-f001:**
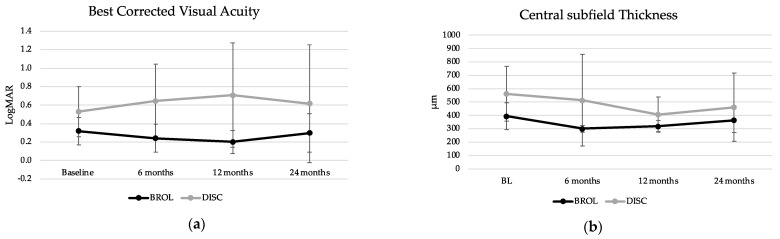
Evolution of best-corrected visual acuity (BCVA) and central subfield thickness (CST) in the case series over the course of two years comparing patients continuing brolucizumab (BROL, n = 5) and patients discontinuing the treatment (DISC, n = 7). (**a**) BCVA in LogMAR and standard deviations. No significant differences between the groups and from baseline to later time points were detected. (**b**) Central subfield thickness (CST, in µm) and standard deviations of the included eyes. No significant differences between groups and from baseline to later time points were detected.

**Figure 2 life-13-00814-f002:**
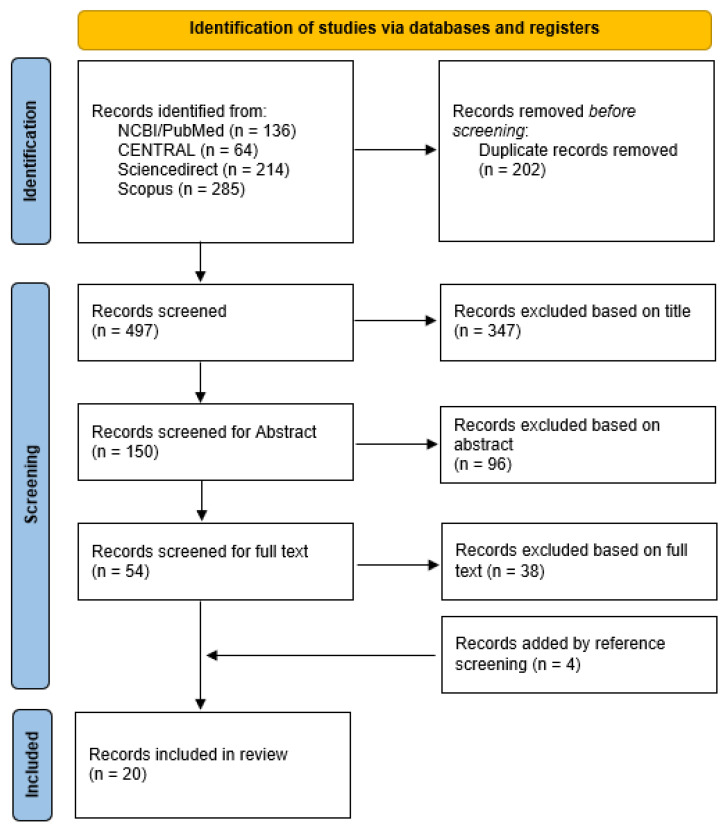
PRISMA search flow diagram presenting the systematic review of the literature for pretreated eyes with neovascular age-related macular degeneration that were switched to intravitreal brolucizumab.

**Figure 3 life-13-00814-f003:**
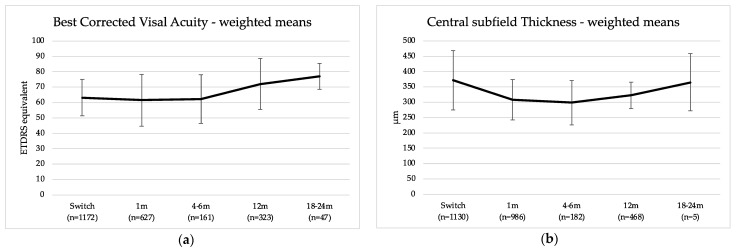
(**a**) Best-corrected visual acuity (BCVA, in early treatment of diabetic retinopathy letters (ETDRS equivalent) converted if necessary) and (**b**) central subfield thickness (CST, in µm) after switch from another anti-VEGF agent to brolucizumab in pretreated neovascular age-related macular degeneration. Weighted mean values from our analysis illustrate point estimates at time of switch (Switch), approximately one month after the first brolucizumab injection (1 m), four to six months after switch (4–6 m, pooled sample), one year after switch (12 m), and 18 to 24 months after switch (18–24 m, pooled sample).

**Table 1 life-13-00814-t001:** Results of meta-analytic calculations over time. N, number of eyes; k_S_, number of samples; point estimate, effect size Cohen’s d; SE, standard error; CI, confidence interval.

	N	k_S_	Point Estimate	SE	95% CI	*p*
BCVA						
BL—1 month	575	11	0.132	0.059	0.016 to 0.248	0.025
BL—4 to 6 months	161	7	0.295	0.108	0.083 to 0.506	0.006
BL—12 months	403	4	0.178	0.136	−0.089 to 0.445	0.191
BL—18 to 24 months	47	3	0.329	0.206	−0.075 to 0.734	0.110
CST						
BL—1 month	575	11	−0.815	0.208	−1.224 to −0.407	<0.001
BL—4 to 6 months	182	8	−1.027	0.265	−1.547 to −0.507	<0.001
BL—12 months	171	4	−1.800	0.639	−3.052 to −0.548	0.005
BL—18 to 24 months	5	1	−0.601	0.598	−1.773 to 0.572	0.315

**Table 2 life-13-00814-t002:** Table representing the peer-reviewed publications of eyes that were switched from another anti-VEGF agent to brolucizumab in neovascular age-related macular degeneration and included in this systematic review and meta-analysis: First author and year represent the cohort and contain data from subsequent publications of the same cohort in two cases; Date, date of publication; Type, type of study; k_s_, separately reported subgroups within a paper; Afl, eyes switched from aflibercept; Bev, eyes switched from Bevacizumab; MNV Type 1, macular neovascularization type 1; PCV, polypoidal choroidal vasculopathy; RS, retrospective study; RCT, randomized controlled trial; n, number of eyes at switch to brolucizumab; FU, follow-up time points where data were available for inclusion in this meta-analysis; CxIOI, percentage of intraocular inflammation as a complication after switch to brolucizumab; CxVasc, percentage of retinal vasculitis and/or vascular occlusion; na, not applicable for not reported data.

First AuthorYear	Paper Title	Journal	Date	Type	k_s_	n	FU (Months)	CxIOI	CxVasc(%)
Abdin et al., 2022 [29]	First Year Real Life Experience With Intravitreal Brolucizumab for Treatment of Refractory Neovascular Age-Related Macular Degeneration	*Frontiers in Pharmacology*	2022-05	RS		21	1, 4–6, 12	9.5%	0%
Bilgic et al., 2021 [30]	Real-world experience with brolucizumab in wet age-related macular degeneration: The reba study	*Journal of Clinical Medicine*	2021-06	RS		80	12	0%	1.25%
Book et al., 2022 [31]	Real-life experiences with Brolucizumab in recalcitrant neovascular age-related macular degeneration	*Der Ophthalmologe*	2022-03	RS		21	4–6	9.5%	0%
Bulirsch et al., 2022 [12]	Short-term real-world outcomes following intravitreal brolucizumab for neovascular AMD: SHIFT study	*Br J Ophthalmol*	2022-09	RS		63	1	11.1%	1.6%
Chakraborty et al., 2022 [21]	Initial experience in treating polypoidal choroidal vasculopathy with brolucizumab in Indian eyes—A multicenter retrospective study	*Indian J Ophthalmol*	2022-04	RS		21	4–6	na	na
Enríquez et al., 2021 [15]	Early Experience With Brolucizumab Treatment of Neovascular Age-Related Macular Degeneration	*JAMA Ophthalmol*	2021-04	RS		151	1	8.1%	0.6%
Hussain et al., 2021 [32]	Brolucizumab for persistent macular fluid in neovascular age-related macular degeneration after prior anti-VEGF treatments	*Ther Adv Ophthalmol*	2021-10	RS	Afl	48	1, 4–6	0%	0%
					Bev	10			0%
Sharma et al., 2021 [11]	Brolucizumab—early real-world experience: BREW study	*Eye (Basingstoke)*	2021-07	RS		42	1	0%	0%
Zuccarini et al., 2022 [33]	Anatomical and functional responses to single brolucizumab injection in neovascular age-related macular degeneration patients not responding to antiangiogenics: a case series	*Drug Target Insights*	2022-03	RS		9	1	0%	0%
Boltz et al., 2022 [34]	Brolucizumab for pre-treated patients with choroidal neovascularization and signs of tachyphylaxis to aflibercept and bevacizumab	*Graefes Arch Clin Exp Ophthalmol*	2022-08	RS		36	1	na	na
Awh et al., 2022 [25]	SHORT-TERM OUTCOMES AFTER INTERIM TREATMENT WITH BROLUCIZUMAB: A Retrospective Case Series of a Single Center Experience	*Retina*	2022-05	RS		51	1	22%	1.9%
Khanani et al., 2022 [16]	MERLIN: Phase 3a, Multicenter, Randomized, Double-Masked Trial of Brolucizumab in Participants with Neovascular Age-Related Macular Degeneration and Persistent Retinal Fluid	*Ophthalmology*	2022-09	RCT		316	4–6	9.3%	2.8%
Ota et al., 2022 [35]	Switching from aflibercept to brolucizumab for the treatment of refractory neovascular age-related macular degeneration	*Jpn J Ophthalmol*	2022-05	RS		48	1	14.6%	6.25%
Ueda-Consolvo et al., 2022 [36]	Switching to brolucizumab from aflibercept in age-related macular degeneration with type 1 macular neovascularization and polypoidal choroidal vasculopathy: an 18-month follow-up study	*Graefes Arch Clin Exp Ophthalmol*	2022-08	RS	MNV Type 1	19	18–24	10.3%	2%
					PCV	23	18–24	21.7%	2%
Montesel et al., 2021 [37]	Short-Term Efficacy and Safety Outcomes of Brolucizumab in the Real-Life Clinical Practice	*Frontiers in Pharmacology*	2021-11	RS		19	4–6	11.1%	0%
Giunta et al., 2022 [22]	Early Canadian Real-World Experience with Brolucizumab in Anti-Vascular Endothelial Growth Factor-Experienced Patients with Neovascular Age-Related Macular Degeneration: A Retrospective Chart Review	*Clin Ophthalmol*	2022-08	RS		73	1, 4–6	4.1%	1.4%
Chakraborty et al., 2021 [38] + Chakraborty et al., 2022 [39]	Brolucizumab in Neovascular Age-Related Macular Degeneration—Indian Real-World Experience: The BRAILLE Study	*Clin Ophthalmol*	2021-08	RS		74	1, 12	0%	0%
	Brolucizumab in Neovascular Age-Related Macular Degeneration—Indian Real-World Experience: The BRAILLE Study—Fifty-Two-Week Outcomes	*Clin Ophthalmol*	2022-12					3.66%	0%
Haensli et al., 2021 [18] + Case series Hänsli et al., 2023	Switching to Brolucizumab in Neovascular Age-Related Macular Degeneration Incompletely Responsive to Ranibizumab or Aflibercept: Real-Life 6 Month Outcomes	*J Clin Med*	2021-06	RS		7	4–6, 12, 18–24	16.7%	8.3%
	Presented Case Series	na	2023						8.3%

## Data Availability

The data presented in this study are described in the text and figures. Further details are available on request from the corresponding author.

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
