# Peer review of "Brolucizumab in Pretreated Neovascular Age-Related Macular Degeneration: Case Series, Systematic Review, and Meta-Analysis"

_life, 2023, doi:10.3390/life13030814_

Round 1

Reviewer 1 Report

The authors reported their experience of switching to brolucizumab in patients with nAMD. In addition, the authors performed systematic review and summarized the current knowledge. The data showed that switching to brolucizumab reduced central subfield thickness but did not improve visual acuity. The drug is potentially useful, but physicians are reluctant to use it because of concern for intraocular inflammation. The presented data would help physicians and patients for their appropriate decision making. The reviewer has only a few comments.

1.     Intraocular inflammation is the most serious concern for the drug. The authors only showed percentage of the complication, but severity also matters. The authors should summarize the number of cases with vasculitis, occlusive events, and severe vision loss if possible.

2.     Line 348 A sentence starting from “Our case series” is left unfinished.

3.     Please provide limitation section. While this is a systematic review, collected studies are all retrospective design with various inclusion and exclusion criteria. The result is essentially not well controlled. Reporting bias would also exists even if authors excluded single case reports of complications.

Author Response

1) We added the number of cases with vasculitis and occlusive events in an additional row in table 1. We inform the reviewer, that these data are mostly referring to the entire cohort of the respective paper and cannot be broken down to pre-treated AMD subgroups. To our surprise, severe vision loss was not consistently reported in most papers.

2) The finding from our case series was added to the terminated sentence, the sentence re-worded.

3) A separate limitation section has been included in addition to some aspects already mentioned in the pre-existing discussion section.

Reviewer 2 Report

As I assess the article, I read the suthors previous report (ref. 18). Comparing the present data from the authors’ case series with their previous report, the results shown in the previous report appear to have clear effects of brolucizumab for 6 months. Are the cases in this paper the same as those in the previous report? Was there no significant increase in visual acuities from baseline for 6 months after brolucizumab treatment?

Author Response

Thank you for the attentive review and comparison. Indeed, our cohort showed some significance for reading acuity six months after switch. Because one of the patients suffered an adverse event, the two cohorts compared in this study do not exactly match, because not all of them continued brolucizumab for two years, even if they are from the same baseline cohort.

Reviewer 3 Report

The authors reported a small case series of Brolucizumab in nAMD refractory to previous anti-vascular endothelial growth factor (anti-VEGF) therapy, and systematic reviewed related literature. There are several issues should be addressed.

1. The word “pretreated” has the meaning as preparing for switch in purpose. It will be better to use “previous treated”.

2. All result data is recommended to demonstrate in tables.

3. The line chart is used to show the change of the continuity of the same variable Figure 1 and 2 should present as histogram. Figure 2 did not contain error bars.

4. In Table 1, some study were focus on polypoidal choroidal vasculopathy. It is different from nAMD. It will be better to exclude, or classify and compare these two different diseases.

5. Line 348, the paragraph was incomplete.

Author Response

Thank you for the critical review, which we addressed in our text, describing the methods section with more precision, and improving the results presentation in our graphs and tables.

1) We defined the word pretreated in the methods section to exclude misperception.

2) Tables 1 and 2 have been revised accordingly.

3) We added error bars and cohort sizes of the pooled and weighted continuous data, which is better to understand in a line chart.

4) We addressed the issue of PCV and AMD in the limitation section.

5) The respective paragraph has been completed.